# Development and Validation of the Italian Pregnancy Nutrition Knowledge Questionnaire (ItPreNKQ): A Nutrition Knowledge Questionnaire for Pregnant Italian Women

**DOI:** 10.3390/nu17050901

**Published:** 2025-03-04

**Authors:** Silvia Callegaro, Elena Bertolotti, Christine Tita Kaihura, Andrea Dall’Asta, Francesca Scazzina, Alice Rosi

**Affiliations:** 1Human Nutrition Unit, Department of Food and Drug, University of Parma, 43124 Parma, Italy; silvia.callegaro@unipr.it (S.C.); elena.bertolotti@unipr.it (E.B.); alice.rosi@unipr.it (A.R.); 2Dipartimento Cure Primarie, U.O.C. Salute Donna, Azienda Unità Sanitaria Locale di Parma, 43126 Parma, Italy; chkaihura@ausl.pr.it; 3Obstetrics and Gynecology Unit, Department of Medicine and Surgery, University of Parma, 43126 Parma, Italy; andrea.dallasta@unipr.it

**Keywords:** pregnancy, nutrition knowledge, ItPreNKQ questionnaire, validation

## Abstract

**Background/Objectives**: Maternal nutrition during pregnancy exerts a significant influence on both maternal and foetal health, as well as long-term child development. Despite its importance, adherence to dietary guidelines among pregnant women remains low. The present study aimed to develop and validate the Italian Pregnancy Nutrition Knowledge Questionnaire (ItPreNKQ), based on national dietary guidelines for the pregnant Italian population, assessing its reliability and validity. **Methods**: The ItPreNKQ comprised 15 questions covering key topics on nutrition during pregnancy. The questionnaire was validated through item analysis (difficulty and discrimination indices), construct validity, internal consistency, and reliability tests. **Results**: A total of 145 pregnant Italian women participated in the study. The reliability of the questionnaire was confirmed through a Pearson’s correlation of R = 0.790 and a Cronbach’s alpha of 0.682, indicating strong temporal stability and acceptable internal consistency. Despite good overall performance, the mean knowledge score was 10.6 ± 2.5 out of 15, indicating significant knowledge gaps in specific topics. **Conclusions**: The ItPreNKQ has been demonstrated to be a reliable and valid tool for the assessment of nutrition knowledge among pregnant Italian women. The tool could be used for assessing nutritional knowledge in prenatal education settings and could be administered in future studies aimed at evaluating the impact of nutritional interventions among pregnant women.

## 1. Introduction

The pivotal role of maternal nutrition during pregnancy in the health and well-being of the pregnant woman, pregnancy outcomes, and the long-term health and development of the offspring is well established [1]. Nutritional requirements increase during pregnancy to maintain maternal metabolism and tissue accretion while supporting foetal growth and development. Therefore, unbalanced diets, poor dietary intakes, or deficiencies in key nutrients can adversely affect pregnancy outcomes and neonatal health [2] leading to an increased risk of preeclampsia, gestational diabetes, and excessive gestational weight gain in the mother, as well as preterm birth, low birth weight, and an increased risk of developing chronic diseases in adulthood in the child [3]. Despite the evidence, several studies have found that knowledge of, and consequently adherence to, national dietary guidelines is low [4,5,6,7]. A systematic review of eighteen studies from ten countries worldwide highlighted that pregnant women did not meet the minimum dietary requirements for vegetables and cereals, which negatively affected their intake of folate, iron, and calcium [8]. Similarly, in the pregnant Italian population, critical nutrients that are often deficient in the diet include docosahexaenoic acid (DHA), iron, iodine, calcium, folic acid, and vitamin D [9].

Dietary behaviours during pregnancy are influenced by several environmental, individual, and psychological factors, such as pre-pregnancy body mass index, nutrition knowledge, nausea and vomiting during pregnancy, perceptions of healthy eating, income, marital status, and ethnicity [10]. Nutrition knowledge (NK) is one of numerous determinants of dietary habits and is defined as an understanding of concepts related to diet, nutrition, and health, including the relationship between diet and health and diet and disease, the nutritional content of foods, and the current dietary guidelines and recommendations [11]. Several studies have demonstrated the role of nutrition knowledge in determining dietary intake and how inadequate nutrition knowledge may act as a barrier to the adoption of healthy behaviours and the maintenance of a healthy body weight [12,13,14,15]. Moreover, various behavioural change theories, such as Social Cognitive Theory [16] and the Knowledge, Attitude, and Practice (KAP) model [17], emphasise the role of nutrition knowledge as a key factor in driving behavioural change, improving food choices, and fostering healthy dietary habits. Both theories suggest that knowledge positively influences an individual’s attitude, which in turn shapes practices and behaviours. In the KAP model, nutrition knowledge serves as a fundamental component, acting as the initial step toward developing positive attitudes and ultimately adopting healthier dietary behaviours [17]. It includes an understanding of dietary guidelines, nutrient functions, and the principles of healthy eating, providing the foundation for making informed dietary choices. Similarly, Social Cognitive Theory highlights that nutrition knowledge not only informs individuals about what to eat but also strengthens their self-efficacy, beliefs, and skills necessary to make and sustain healthy dietary decisions [16,18]. Fortunately, pregnancy is considered a “teachable moment”, a naturally occurring life transition or health event that is thought to motivate individuals to spontaneously adopt risk-reducing health behaviours [19]. Therefore, an improvement in NK during this particular life phase could be an important opportunity to establish healthy behaviours in mothers that could be maintained for the rest of their lives and have beneficial consequences for their offspring as well [20,21]. Several studies in the literature have demonstrated that enhancing nutrition knowledge through educational interventions leads to improved food choices among pregnant women, supporting a positive behaviour change [18,22,23,24,25]. Consequently, ascertaining the extent of nutritional knowledge among pregnant women is imperative for the formulation of corrective actions and initiatives, such as nutrition education programmes on dietary guidelines, aimed at enhancing their knowledge of food choices to build strong, healthy behaviours.

Specific questionnaires, appropriately developed and tailored to collect information on NK in target populations, are commonly used. A few NK questionnaires have been validated worldwide to evaluate the level of knowledge in the pregnant population and have been used in both observational and interventional studies [7,26,27,28,29,30,31]. To the best of our knowledge, no NK questionnaire focusing on dietary recommendations for pregnant women has been validated in the Italian population. Therefore, the aim of this study was to develop and validate a nutrition knowledge questionnaire based on the Italian food-based dietary guidelines for the pregnant Italian population.

## 2. Materials and Methods

### 2.1. Development of the Italian Pregnancy Nutrition Knowledge Questionnaire (ItPreNKQ)

The Italian Pregnancy Nutrition Knowledge Questionnaire (ItPreNKQ) was created with the aim of investigating the NK of pregnant Italian women regarding pregnancy-specific dietary and nutritional recommendations. The development of the ItPreNKQ was based on previous nutrition knowledge questionnaires found in the literature [7,26,27,28] and on other scientific sources, such as the Italian food-based dietary guidelines [32]. As reported in Figure 1, this process yielded an initial draft list of 39 questions, which was then revised by a group of experts in nutrition and gynaecology. The final questionnaire comprised 15 items based on Italian dietary recommendations for pregnant women.

The questionnaire was divided in two sections exploring various topics related to nutrition and pregnancy such as weight gain, food safety concerns, intake frequencies of food groups, and the recommended intake, supplementation, and food sources of critical nutrients that are often insufficient in the diet of pregnant Italian women such as folates, omega-3, calcium, and iron [9].

For each item, there was a correct answer, two incorrect answers, and an additional “I don’t know” option to prevent missing data or random responses. In particular, 1 point was given for correct answers, while 0 points were given for incorrect or “I don’t know” responses, resulting in a total score ranging between 0 and 15 points, with higher scores indicating stronger NK.

To validate the ItPreNKQ, item analysis (item difficulty and item discrimination index) and construct validity analysis were conducted, and internal consistency and test–rest reliability were assessed.

### 2.2. Participants and Questionnaire Administration

Participants were recruited by sharing the initiative on major communication and social media platforms, as well as by posting and distributing flyers with a QR code linking to the questionnaire in key areas of interest for pregnant women (counselling centres, prenatal classes, and public and private gynaecology clinics) from June 2024. The inclusion criteria were being pregnant, nulliparous, over the age of eighteen, of Italian nationality, and residing in Italy. The study protocol was approved by the Research Ethics Board (REB) of the University of Parma (protocol number 0145719, 11 June 2024).

The questionnaire was administered as an online survey through EUSurvey, a dedicated European platform for the creation and distribution of online surveys and forms [33]. Data were collected anonymously by selecting the “Anonymous Survey Mode” option, thereby ensuring that the platform did not store any of the participants’ personal data, including IP addresses. To ensure the maintenance of anonymity, participants were identified by alphanumeric codes, and the data collected were treated in aggregate form, with no reference to the subjects who completed the ItPreNKQ. Informed consent was obtained at the beginning of the questionnaire; once the participant selected the “I agree” option, they could proceed to complete the questionnaire. At the initial administration, participants were asked to answer a series of screening questions aiming to identify subjects meeting the inclusion criteria. In addition, a set of socio-demographic questions was administered for a complete classification of the participants. In this section, participants were asked to provide information on their pregnancy status and trimester, age, nationality, geographical area of residence in Italy (Northwest, Northeast, Central, South, or Islands), education level (pre-graduated, graduated, or post-graduated), occupation (unemployed, part-time, full-time, student, or maternity leave), and whether they had a background in nutrition or health (having attended a university course on topics related to food, dietetics, and/or nutrition). An e-mail address was also collected to contact participants for the second administration of the questionnaire. The ItPreNKQ was administered to each participant, under identical conditions, on two separate occasions (Figure 1). The second administration was carried out two weeks later, a timeframe which is commonly considered both long enough to forget exact answers and short enough that no new information would be learned [34]. A reminder was sent to participants by e-mail to remind them of the date on which they had to complete the ItPreNKQ on the second occasion.

### 2.3. Sample Size and Statistical Analysis

According to the existing literature, the required sample size for questionnaire validation should be at least six to ten times the number of questions [34]. Given that the NK questionnaire contained 15 items, the total number of participants required for validation was estimated to range between 90 and 150.

A descriptive analysis was performed to explore participants’ socio-demographic characteristics. Continuous variables were expressed as mean and standard deviation, while categorical variables were reported as frequencies (%) and absolute values.

Item analysis was conducted by calculating the item difficulty index and the item discrimination index for each question. The item difficulty index, defined as the percentage of correct responses for each question, was evaluated using cut-off values set between 0.1 and 0.9 [34,35]. This range excludes items with a proportion of correct answers higher than 90% or lower than 10%, as such items are not informative in assessing subjects’ knowledge. Although cut-off values of 0.2 and 0.8 are generally used [36], a more conservative approach was chosen to retain items that provided significant contributions to the overall knowledge score, as supported by other studies [35,37,38,39]. The item discrimination index was determined using a point-biserial correlation between each question’s score and the total score, with a minimum value of 0.2 being considered adequate to indicate an acceptable correlation [34,36].

Construct validity was assessed using the Welch t-test for independent samples, comparing NK scores between subjects with and without a nutrition background, based on responses from the initial administration of the questionnaire (t1).

The internal consistency of the total nutritional knowledge score was evaluated using Cronbach’s alpha, considering that all items included more than two response options [34]. Cronbach’s alpha values greater than 0.6 were considered acceptable [34,40].

Additionally, Pearson’s correlation and the intraclass correlation coefficient (ICC) were used to assess temporal stability (test–retest reliability) by comparing the total scores from the initial administration (t1) with those from the follow-up administration (t2).

All statistical analyses were performed using Statistical Package for Social Science (IBM SPSS Statistics, version 29.0, IBM Corp., Armonk, NY, USA) and the significance was set at *p* < 0.05.

## 3. Results

### 3.1. Participants’ Characteristics

A total of 145 pregnant Italian women in their first pregnancy were enrolled in the study and included in the final analysis. Their socio-demographic characteristics are detailed in Table 1. The sample had a mean age of 33 ± 4 years, and the majority of participants (66%) were in the third trimester of pregnancy, while only 7% were in the first trimester. Most women (84%) resided in the north of Italy. Regarding education, a great number of the subjects held a university degree (64%), while only 19% reported having a nutritional background (NB). At the time of the questionnaire administration, most of the women were full-time employees (48%) or currently on maternity leave (41%).

### 3.2. Item Analysis

The 15 questions of the questionnaire and their respective item difficulty and item discrimination indexes are presented in Table 2.

Item difficulty ranged between 0.12 and 0.99, with two questions (numbers 3 and 9) having an item difficulty index above 0.9. However, none of the questions were correctly answered by all participants, so they were retained due to the importance of their topic (folic acid supplementation and alcohol consumption). The item discrimination index ranged from 0.191 to 0.643, with only one item (number 13) scoring below 0.2. This indicates a generally good score correlation between the score of each question and the total score of the questionnaire.

### 3.3. Questionnaire Validity and Reliability

The mean NK total score at baseline (t1) was 10.6 ± 2.5 out of 15, indicating a medium–high NK (Table 3). The mean score was 11.4 ± 2.4 for participants with a nutritional background and 10.4 ± 2.5 for those without it, although this difference was not statistically significant. A total of 104 participants completed the second administration. Their average NK score was 11.1 ± 2.2 at baseline and increased to 11.5 ± 2.1 at the second administration. With regard to the reliability analysis, a good time reliability was observed (R = 0.790, 95% CI: 0.704–0.853; interclass correlation coefficient ICC = 0.678, 95% CI: 0.651–0.704), indicating strong stability of the questionnaire over time (Table 3). Similarly, the internal consistency of the ItPreNKQ was satisfactory, with a Cronbach’s alpha of 0.682 (Table 3). The overall reliability of the NK questionnaire remained consistent when individual items were removed. Therefore, all questions were retained in the final version of the ItPreNKQ.

## 4. Discussion

Nutrition knowledge is a critical factor influencing dietary behaviours. Given the importance of healthy eating habits during pregnancy to improve the health of both mothers and their offspring, enhancing nutrition knowledge among pregnant women could exert a favourable impact on pregnancy outcomes. To the best of our knowledge, this is the first time that a questionnaire has been developed and validated for the Italian population of pregnant women to explore their knowledge of key dietary recommendations during pregnancy. Indeed, a number of nutrition knowledge questionnaires have been validated for specific groups of the Italian population [37,41,42,43,44], but none have been developed taking into account specific knowledge related to nutrition and pregnancy. Examining the literature beyond Italy, some questionnaires designed for pregnant women were found, but they are based on the specific nutritional needs and dietary recommendations of a particular country or population [7,26,27,28,29,30,31]. This underscores the necessity for a tool that can comprehensively assess the specific nutritional knowledge of pregnant Italian women, considering the unique characteristics of Italian dietary habits and gastronomic traditions, as well as any endemic micronutrient deficiencies present within the Italian population [40]. Despite the initial cut-off of 0.1–0.9 defined for the item difficulty index, two questions exceeded this range with values greater than 0.9. These questions pertained to folic acid supplementation and alcohol consumption, two topics considered to be of paramount importance by the expert panel. It is highly probable that, because they are of such importance, these topics are already well known among pregnant Italian women. Additionally, the exclusion of these questions would not have improved the questionnaire’s reliability.

Regarding the discrimination index, only question 13, related to calcium content in foods, had a borderline value slightly below 0.2 (R = 0.191). The expert panel decided to retain the question in the final questionnaire as it pertains to food sources of calcium, a key topic given that calcium requirements increase during pregnancy. The remaining questions demonstrated moderate-to-good correlation values (R > 0.2), indicating the capacity of items to correctly discriminate between people with different levels of knowledge. So, all 15 items were retained in the final version of the questionnaire as they addressed key topics related to nutrition knowledge during pregnancy.

The results of the construct validity analysis suggest that participants with a nutritional background generally exhibit a higher score compared to those without such a background, although this difference was not statistically significant. This suggests that the questionnaire has a modest capacity to differentiate between respondents with higher or lower expected nutrition knowledge based on their educational background. However, the limited sample size of subjects with a nutritional background in comparison to subjects without it may have influenced the statistical significance.

The temporal stability of the ItPreNKQ was evaluated using the test–retest method, wherein the questionnaire was administered twice at different times, demonstrating good temporal stability (R = 0.790).

The tool also showed adequate overall internal consistency and reliability. Although a Cronbach’s alpha value of 0.7 [34,40,45] or higher is widely accepted as the standard for adequate internal consistency, a Cronbach’s alpha value greater than 0.6 was considered acceptable, which is consistent with previous studies [40,45,46,47].

In general, the participants demonstrated a good level of knowledge on key topics such as the increased need for essential micronutrients during pregnancy (e.g., folic acid, iron, and calcium), the risks associated with commencing pregnancy in a state of overweight, the consumption of alcohol during pregnancy, the importance of omega-3 intake for foetal development, and aspects of food safety. However, lower levels of knowledge were observed on more specific topics, including the reasons why nutrient requirements increase during gestation, food sources of key micronutrients, particularly calcium, methods to enhance iron absorption, appropriate coffee intake, and recommended weight gain during pregnancy. These findings are consistent with the results of other studies conducted worldwide [7,26,28,48]. For instance, Lucas et al. [49] investigated the knowledge and use of iodine supplements in a sample of pregnant Australian women. Despite 94% of participants being aware of the health risks associated with iodine deficiency, there was limited knowledge regarding food sources of iodine and the specific health problems caused by its deficiency. A similar lack of knowledge was found in a recent study conducted in Lebanon, which revealed that most participants were unaware of the crucial role of nutrients such as iodine and omega-3. Moreover, only a small percentage were able to correctly identify iron-rich foods or sources of omega-3 [50]. The results obtained in this study, along with evidence from other studies in the literature, highlight that nutritional knowledge of pregnant women often remains superficial. While basic concepts are generally understood, more specific and critical topics tend to be overlooked. The present study suggests that the observed knowledge gap may be partly explained by the fact that many healthcare professionals such as midwives and gynaecologists typically provide information on critical aspects of prenatal nutrition, such as the importance of folic acid supplementation and the need to avoid alcohol. However, more detailed and comprehensive nutritional education is often lacking. A 2024 review [20] highlighted that levels of adherence to dietary guidelines are low globally, primarily due to persistent knowledge gaps. For instance, while many women understand the importance of folic acid supplementation, they often lack knowledge about its dietary sources or its recommended dosage. Moreover, many pregnant women report dissatisfaction with the nutrition education received during prenatal visits, describing it as too general, impractical, and insufficient [20]. Furthermore, the information provided is often limited to topics such as the prevention of foodborne infections or weight control, while critical areas such as the risks of nutritional deficiencies and the dietary sources of essential nutrients [51] are often overlooked. Additionally, they feel that nutritional information is provided too late, expressing a preference for receiving such guidance in the first trimester of pregnancy or even prior to conception [20]. These findings underscore the need for targeted nutrition education interventions specifically designed for pregnant women. Integrating structured nutrition education into prenatal care, for instance, by incorporating it into pre-partum classes, could address this gap. The provision of such interventions before or at the beginning of the pregnancy would facilitate the acquisition of comprehensive knowledge by pregnant women, enabling them to make informed dietary choices and supporting healthy pregnancy outcomes. Nutrition education and counselling activities have demonstrated their importance during pregnancy, showing improvements in dietary behaviours, adherence to micronutrient supplementation, and both maternal and neonatal health outcomes [18,23,24,25,52,53,54]. When effectively implemented, these interventions could play a pivotal role in bridging the gap between awareness and action, promoting the well-being of both mothers and their children.

It is important to acknowledge some limitations and strengths of the study when evaluating its findings. Among the limitations, although the questionnaire and its validation were open to women from all over Italy, the sample was geographically unbalanced, with the majority of participants residing in Northern Italy, limiting the generalisability of the results. Expanding the recruitment to other Italian regions would have enhanced the generalisability of the results for the whole pregnant Italian population. Another limitation to generalisability is the enrolment setting: although the initiative was promoted in both community and private settings, most of the women enrolled attended public health facilities. This may have introduced selection bias, limiting the representation of women who access private health care and have access to different nutritional information than women who use public health care services. Future studies should aim to diversify recruitment by including a higher number of participants from private healthcare providers, non-medical community organisations, and online platforms, as well as ensuring better geographic representation across all Italian regions. Moreover, while the study found that participants with a nutritional background scored higher, the difference was not statistically significant (*p* = 0.055). This may be due to the small sample size of individuals with a nutritional background. A sample that includes a more balanced representation of individuals with and without a nutritional background could provide a clearer validation of construct validity and better assess whether nutrition education significantly influences knowledge levels. Furthermore, most of the participants were in their third trimester of pregnancy and had likely already attended pre-partum classes, where they may have acquired information about nutrition in pregnancy. Potential confounding variables such as dietary habits and educational levels were not analysed in relation to nutrition knowledge levels. Future studies should incorporate more detailed statistical analyses to assess how these factors may influence the outcomes, ensuring a more comprehensive understanding of the determinants of nutrition knowledge in pregnant women.

On the contrary, one of the main strengths of this study is that the ItPreNKQ is based on the Italian food-based dietary guidelines [32], comprehensively covering several aspects of food safety and nutrition for pregnant Italian women. In addition, the decision to include only nulliparous women aimed to avoid any potential bias that may be attributable to prior knowledge from previous pregnancies.

## 5. Conclusions

In conclusion, the ItPreNKQ is a reliable, valid, and easy-to-use tool for assessing the nutritional knowledge of pregnant Italian women. Future studies should investigate its use in prenatal educational interventions to evaluate changes in the knowledge of the pregnant women or within prenatal classes to evaluate participants’ level of knowledge and identify knowledge gaps that need to be addressed. To minimise selection bias in future studies and improve the representativeness of the sample, a more balanced recruitment approach should be implemented. This could include the collaboration with private healthcare providers and the use of pregnancy-related social media communities to reach participants, also through multicentric studies, to increase the generalisability of the tool. Last, the assessment of potential confounding variables, such as dietary habits, could improve the understanding of the results of the questionnaire.

## Figures and Tables

**Figure 1 nutrients-17-00901-f001:**
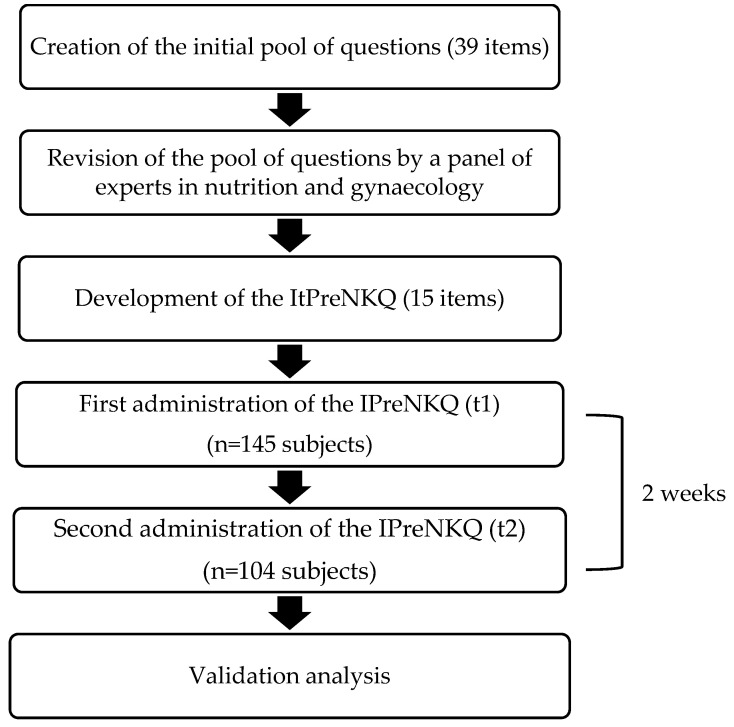
Flowchart of the ItPreNKQ questionnaire development and validation process.

**Table 1 nutrients-17-00901-t001:** Participants’ characteristics (*n* = 145).

(*n* = 145)	
Age (Years)	32.7 ± 4.2
Trimester of pregnancy	
First trimester (0–12 weeks)	10 (6.9%)
Second trimester (13–27 weeks)	39 (26.9%)
Third trimester (28–40 weeks)	96 (66.2%)
Italian geographical area	
Northwest (Aosta Valley, Piedmont, Liguria, Lombardy)	15 (10.3%)
Northeast (Emilia–Romagna, Veneto, Trentino–South Tyrol, Friulia Venezia Giulia)	107 (73.8%)
Central (Tuscany, Marche, Umbria, Lazio)	12 (8.3%)
South (Abruzzo, Molise, Campani, Puglia, Basilicata, Calabria)	9 (6.2%)
Islands (Sicily and Sardinia)	2 (1.4%)
Educational level	
Pre-graduated	36 (24.8%)
Graduated	93 (64.2%)
Post-graduated	16 (11.0%)
Employment	
Unemployed	3 (2.1%)
Student	1 (0.7%)
Full-time employee	69 (47.6%)
Part-time employee	13 (9.0%)
Employed currently on maternity leave	59 (40.7%)
Nutritional Background	
Yes	28 (19.3%)
No	117 (80.7%)

Data are expressed as mean ± SD or as an absolute number (frequency).

**Table 2 nutrients-17-00901-t002:** Item analysis for the 15 items of the NK questionnaire.

Item	I Don’t Know Answers (%)	Item Difficulty (Correct Answers)	Item Discrimination (R Value)
1—For a normal-weight woman, what is the recommended weight gain during pregnancy?	1.4%	0.23	0.317
2—Starting the pregnancy overweight:	6.2%	0.88	0.467
3—When is it recommended to start folic acid supplementation?	0.0%	0.92	0.231
4—Indicate which of the following foods is a good source of folic acid	18.6%	0.76	0.398
5—Why is folic acid supplementation recommended during pregnancy?	2.8%	0.69	0.643
6—During pregnancy, daily iron recommend intake:	2.8%	0.88	0.301
7—Which of the following strategies allows the body to increase iron absorption?	23.4%	0.64	0.592
8—Coffee consumption during pregnancy:	1.4%	0.74	0.423
9—Alcohol consumption during pregnancy:	0.0%	0.99	0.266
10—Why does the need for long-chain omega-3 fatty acids (DHA and EPA) increase during pregnancy?	25.5%	0.70	0.553
11—To cover Omega-3 requirements in pregnancy is recommended:	11.7%	0.84	0.459
12—Why does the daily calcium requirements increase during pregnancy?	7.6%	0.88	0.509
13—Indicate which of the following foods is a good source of calcium:	26.2%	0.12	0.191
14—How many servings of fruits and vegetables are recommended during pregnancy?	9.0%	0.49	0.449
15—Which of the following fish should be avoided during pregnancy because it is rich in mercury?	9.7%	0.83	0.515

**Table 3 nutrients-17-00901-t003:** Construct validity, internal consistency, and test–retest reliability.

	Baseline Score (t1) (*n* = 145)	Nutritional Background (*n* = 28)	No Nutritional Background (*n* = 117)	*p*-Value *	Baseline Score (t1) *n* = 104	Follow-Up Score (t2) *n* = 104	Correlation	ICC	Cronbach’s α
Score Range	Mean ± SD	Mean ± SD	Mean ± SD		Mean ± SD	Mean ± SD	R (95% CI)	ICC (95% CI)	
0–15	10.6 ± 2.5	11.4 ± 2.4	10.4 ± 2.5	0.055	11.1 ± 2.2	11.5 ± 2.1	0.790 (0.704, 0.853)	0.678 (0.651, 0.704)	0.682

SD, standard deviation; ICC, intraclass correlation coefficient; CI, confidence interval; * Welch *t*-test for independent samples. The *p*-value indicates the statistical significance of the difference between groups with a nutritional background and with no nutritional background.

## Data Availability

The raw data supporting the conclusions of this article will be made available by the authors upon request.

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
