# Peer review of "Development and Validation of the Italian Pregnancy Nutrition Knowledge Questionnaire (ItPreNKQ): A Nutrition Knowledge Questionnaire for Pregnant Italian Women"

_nutrients, 2025, doi:10.3390/nu17050901_

Round 1

Reviewer 1 Report

Comments and Suggestions for Authors

The study introduces the Italian Pregnancy Nutrition Knowledge Questionnaire (ItPreNKQ), which is the first validated questionnaire specifically designed for assessing nutrition knowledge among pregnant women in Italy. This is a novel contribution to the field, as no prior validated tools exist for this population. Based on the article, the authors could consider the following methodological improvements and additional controls:

1.The study population is predominantly from Northern Italy (84%), which may not accurately reflect the entire Italian pregnant population. Expanding recruitment to other regions would enhance generalizability.

2.Most participants were recruited through public health facilities, potentially introducing selection bias. Including participants from private healthcare settings and non-medical community settings would improve representation.

3.The study found that participants with a nutritional background scored higher, but the difference was not statistically significant. A larger sample size of individuals with and without formal nutrition education could provide a clearer validation of construct validity.

4.Variables such as education level, and dietary habits should be more explicitly analyzed as potential confounders in nutrition knowledge levels.

By addressing these methodological limitations and incorporating further controls, the study could improve the robustness of its findings and ensure the Italian Pregnancy Nutrition Knowledge Questionnaire (ItPreNKQ) is a reliable and valid tool across diverse populations.

Reviewer 2 Report

Comments and Suggestions for Authors

The manuscript is organized, but some issues need to be corrected before the manuscript can be published 

Abstract:

 Line 28 29: It can be used in prenatal education programs to evaluate the effectiveness of interventions and identify knowledge gaps that need to be addressed for pregnant women due to their special nutritional needs—this phrase does not correlate with the results.

Keywords: - questionnaire please add  ItPreNKQ

Introduction:  short but OK

Materials and methods: A flow chart must be added  (the number of patients who resulted at each study step ….  ).   What is the power of the study? How many p[patients are needed for the study to be statistically significant?

Results: Tables 1 and 2 can be converted to a graph. Regarding the correlation between the variables – why wasn’t an ANOVA test used ANOVA test results with F statistics and P-Value? – or / Turkey test? Post-Hoc Analysis (Tukey HSD Test)(Honestly Significant Difference) test ? is there a control group for the study?

Discussions: Two short. A great deal of time must be spent commenting on the results and comparing them to the published data.

Conclusions: lines 314-315  The questionnaire could be used in future educational interventions to assess their effectiveness or within prenatal classes to evaluate participants’ level of knowledge and identify knowledge gaps that need to be addressed. Nut supported by the results could be added to the discussion part.

Round 2

Reviewer 2 Report

Comments and Suggestions for Authors

The authors have reponded to all of the comments. The manuscriptr has been improved.